# Circular RNAs in Hedgehog Signaling Activation and Hedgehog-Mediated Medulloblastoma Tumors

**DOI:** 10.3390/cancers13205138

**Published:** 2021-10-13

**Authors:** Ani Azatyan, Shasha Zhang, Anna Darabi, Peter Siesjö, Ting Wang, Peter G. Zaphiropoulos

**Affiliations:** 1Department of Biosciences and Nutrition, Karolinska Institutet, 14183 Huddinge, Sweden; ani.azatyan@ki.se; 2Department of Pharmacy, Sichuan Cancer Hospital & Institution, School of Medicine, University of Electronic Science and Technology of China, Chengdu 610041, China; 201922130414@std.uestc.edu.cn; 3Department of Clinical Sciences Lund, Neurosurgery, Faculty of Medicine, Lund University, 22185 Lund, Sweden; anna.darabi@med.lu.se; 4Department of Clinical Sciences Lund, Neurosurgery, Skåne University Hospital, Lund University, 22185 Lund, Sweden; peter.siesjo@med.lu.se

**Keywords:** Hedgehog pathway, circRNA, back-spliced junction, RNA-seq, cerebellar tumor

## Abstract

**Simple Summary:**

Here the expression profile of circular RNAs in Hedgehog signaling-dependent cell lines and medulloblastoma cells was interrogated. Using stringent criteria, a reduced expression of seven circular RNAs in Hedgehog-dependent medulloblastoma versus cerebellum was clearly established. Depletion and/or overexpression of these deregulated RNA circles in two medulloblastoma cell lines revealed minimal effects in cellular proliferation based on two independent assays. These findings highlight the complexity of gene expression outcomes and the possibility that gene products may not necessarily have an obvious phenotypic impact on the cellular context where they are present. It is not inconceivable that a substantial number of differentially expressed circular RNAs may represent “passenger molecules” with little impact on a cell, reflecting the stochasticity of the gene expression and splicing processes.

**Abstract:**

Within the past decade, circular RNAs have largely emerged as novel regulators of human biology, including brain function and cancer development. On the other hand, the Hedgehog pathway has established roles in regulating biological processes, including tumorigenesis. Here, the circular RNA transcriptome, in the context of Hedgehog signaling activation of medulloblastoma Daoy and human embryonic palatal mesenchyme HEPM cells, was determined. In total, 29 out of the 30 selected circular RNAs were validated by Sanger sequencing, with some regulated to a limited extent by Hedgehog signaling. Interestingly, back-spliced junctions, the marker of exonic RNA circles, were also identified at a low frequency within poly (A) mRNAs, reflecting exon repetition events. Thirteen circular RNAs had reduced expression in human medulloblastoma tumors in comparison to normal cerebellum. For seven out of these thirteen RNA circles, the linear mRNAs originating from the same genes did not exhibit a reduced expression. Depletion and/or overexpression of these seven circular RNAs minimally affected medulloblastoma cell proliferation. These findings highlight that differential expression of a gene product may not necessarily elicit an obvious phenotypic impact. Consequently, further analysis is required to determine the possible subtle contributions to the development of this cerebellar tumor.

## 1. Introduction

Hedgehog signaling has major roles in normal development and homeostasis and its continuous activation is implicated in tumor initiation and growth, including basal cell carcinoma, medulloblastoma, rhabdomyosarcoma, breast, colon, pancreatic, and lung cancers [1,2,3,4,5,6]. Medulloblastoma is a highly aggressive brain tumor, with the highest occurrence among children. Medulloblastomas exhibit considerable genomic heterogeneity, with the Sonic Hedgehog (SHH) subtype being one of the largest, comprising about 30% of all medulloblastoma cases [7,8]. SHH plays critical roles in controlling the development of the cerebellum, while amplifications and mutations of the Hedgehog pathway components (GLI2, GLI1, PTCH1, SMO, and SUFU) are common drivers of SHH medulloblastomas [9,10,11]. As conventional therapies for this cerebellar tumor are implemented through invasive methods, with the current survival rate of medulloblastoma patients reaching plateau levels [12], a better understanding of the biology of medulloblastoma is likely to contribute to the development of novel treatment strategies.

Circular RNAs (circRNAs) in eukaryotes were identified more than twenty years ago [13,14,15] but have largely been ignored as non-functional by-products. However, in recent years there is a renewed interest in RNA circles stemming from their emerging roles in human biology [16,17,18,19]. circRNAs are generated by an alternative splicing event, when the 3′ end of an exon is back-sliced to the 5′ end of its upstream exon. This is facilitated by inverted repeats in the introns that flank the back-spliced exons [20] and may be linked to a lariat-mediated skipping of the circularized exons [21]. Covalently closed RNA circles lack the CAP structure and the polyadenylated poly (A) tail of mRNAs and are characterized by an increased stability. circRNAs have highly specific tissue expression patterns, with preferential abundance in brain tissues [16,17,22] and were demonstrated to act as oncogenes or tumor suppressors in colorectal, hepatic, prostate, and bladder cancers [23,24,25,26].

In this study, we address the role of circRNAs in the context of Hedgehog signaling activation and Hedgehog-driven medulloblastoma tumors. Using modified RNA-seq protocols and implementing several selection criteria, 29 out of the 30 selected circRNA were validated with Sanger sequencing. Seven of these circRNAs were downregulated in medulloblastoma compared to normal cerebellum, while their corresponding linear mRNAs remained unchanged. This underscores the differential splicing mechanisms acting on these RNA circles, suggesting possible implications in the context of cerebellar tumor growth.

## 2. Materials and Methods

### 2.1. Cell Culture

The Daoy medulloblastoma cells were a kind gift of F. Aberger (University of Salzburg, Salzburg, Austria), the UW-228 medulloblastoma cells were a kind gift of J. I. Johnsen (Karolinska Institutet, Solna, Sweden), and the HEPM (CRL-1486) human embryonic palatal mesenchyme cell line was purchased from ATCC (American Type Culture Collection, Manassas, VA, USA). Cell lines were assessed for mycoplasma contamination (LT07-218, Lonza, MycoAlert™mycoplasma detection kit, Basel, Switzerland). Daoy and HEPM cells were cultured in EMEM medium with L-glutamine (Lonza, Basel, Switzerland) and UW-228 cells in DMEM/F12 (1:1) medium with L-glutamine (11320033, Gibco, Thermo Fisher Scientific, Waltham, MA, USA), supplemented with 10% fetal bovine serum (FBS) (A31608-02, Gibco) and 100 IU/mL penicillin/streptomycin (15140-122, Gibco, Thermo Fisher Scientific, Waltham, MA, USA) and maintained in a 5% CO_2_ humidified incubator at 37 °C.

### 2.2. Drug Treatments

The Smoothened agonist (SAG, 566660) was purchased from Merck (Darmstadt, Germany) and the recombinant human Sonic Hedgehog N-terminus ligand (SHH-C24II, 185-SH) was purchased from R&D Systems (Minneapolis, MN, USA). SAG was dissolved and diluted in DMSO, and SHH was dissolved and diluted in PBS with 0.1% BSA.

Daoy cells were seeded in 6-well plates at ≈30% confluency in complete medium, then starved in Optimem medium (11058-021, Gibco, Thermo Fisher Scientific, Waltham, MA, USA) for 24 h and treated with either 200 nM SAG in EMEM medium supplemented with 0.5% FBS for 72 h or with 400 ng/mL SHH in Optimem for 72 h. From each treatment condition two separate wells were combined and processed further for RNA isolation. Similarly, for the control (Ctr) treatments, cells were either starved in Optimem for 24 h and mock-treated with SAG solvent (DMSO) in EMEM medium supplemented with 0.5% FBS for 72 h (SAG_Ctr) or with SHH solvent (0.1% BSA in PBS) in Optimem for 72 h (SHH_Ctr).

HEPM cells were seeded in 10 cm cell culture dishes at ≈40% confluency in complete medium, then starved in Optimem for 24 h and treated with either 200 nM SAG or 400 ng/mL SHH in Optimem medium for 72 h or with DMSO + 0.1% BSA in PBS as the control treatment.

### 2.3. Human Normal Brain Cerebellum and Medulloblastoma Tumor Samples

The three normal cerebellar RNA samples were purchased from BioChain, Newark, CA, USA (R1234039-50/Lot:B910013 and R1234039-50/Lot:C409088), and TakaraBio, Kusatsu, Shiga, Japan (636535/Lot:1802037). The first two were from single individuals, while the third sample was a pool from three individuals. The collection and subsequent analyses of the patient-derived SHH subtype medulloblastoma tumor tissue sample were approved by The Swedish Ethical Review Authority of Lund University (Dnr 2018/37) and collected under Södra sjukvårdsregionens tumörbiobank (Appendix A).

### 2.4. RNA Isolation

Total RNA from cell lines or the medulloblastoma tumor sample was isolated with the E.Z.N.A total RNA kit I (R6834-02) or total RNA midi kit (R6664-02) (Omega Bio-tek, Norcross, GA, USA) according to the manufacturer’s instructions. For the RNA isolation from the nitrogen snap-frozen tumor tissue sample, a small piece (10 mg) was homogenized by Qiagen TissueRuptor (Hilden, Germany). On-column DNA digestion was performed with 40 KU (R6834-02 kit) and 60 KU (R6664-02 kit) of DNase (79254, Qiagen, Hilden, Germany).

### 2.5. circRNA Enrichment and RNA-Seq Library Preparations

Before proceeding to library preparation, total RNA was assessed for RNA integrity with the RNA ScreenTape System (G2964AA, Agilent Technologies, Santa Clara, CA, USA) using the TapeStation 2200 (Agilent Technologies, Santa Clara, CA, USA).

To ensure unbiased representation of both linear poly (A) mRNA and circRNA, the total RNA preparations were divided into two fractions and subjected to different pipelines optimized for detection of either linear poly (A) or circRNAs.

(a) Poly (A) enriched fraction: ≈100 ng of total RNA was used to generate stranded mRNA libraries with the TruSeq^®^Stranded mRNA Low Throughput Sample Preparation kit according to the manufacturer’s instructions (15031047, Illumina, San Diego, CA, USA). This protocol involves a poly (A) selection/enrichment and ribosomal RNA depletion step.

(b) circRNA enriched fraction: 5 μg of total RNA was used. First, ribosomal RNA was depleted using the RiboMinusTM Eukaryote System V2 (A15026, Ambion, Thermo Fisher Scientific, Waltham, MA, USA), followed by digestion with 5U RNase R for 1 h to degrade linear RNAs and purification with the RNeasy MinElute Cleanup Kit (74204, Qiagen, Hilden, Germany). The concentration and quality of the enriched circRNA RiboMinus/RNase R preparations were assessed with an Agilent 2100 Bioanalyzer (Agilent Technologies, Santa Clara, CA, USA). A total of 4.4 ng (for Daoy cells) or 1 ng (for HEPM cells) of RiboMinus/RNase R treated RNA was used for cDNA library generation. Random hexamer primers and Superscript III reverse transcriptase (18080055, Invitrogen, Thermo Fisher Scientific, Waltham, MA, USA) were used for 1st strand synthesis and the stranded mRNA library was generated according to the TruSeq^®^Stranded mRNA Low Throughput Sample Preparation protocol (15031047, Illumina, San Diego, CA, USA) as described above.

The concentration and quality of the cDNA libraries were assessed with Qubit (Life Technologies, Thermo Fisher Scientific, Waltham, MA, USA) and TapeStation, respectively. The cDNA libraries were normalized and pooled for sequencing on the Illumina NextSeq 550 platform using the Illumina NextSeq 500/550 High Output v2 kit (75 cycles; single read). Basecalling and de-multiplexing analyses were performed using CASAVA software with default settings generating Fastq files for further downstream mapping and analysis.

### 2.6. RNA-Seq Data Analysis

(1) mRNA detection: reads from the poly (A) enriched fraction were aligned to the Gencode GRCh38 reference genome assembly using the TopHat (v2.5.2) aligner. Counts were assigned to genes using featureCounts (v1.5.1).

(2) circRNA detection: reads from the circRNA enriched fraction were analyzed with the CIRCexplorer2 characterization pipeline in order to identify back spliced junctions and aligned to Gencode GRCh38 using CIRCexplorer2/TopHat-Fusion [27] using the default parameters. Back-spliced junctions are identified as reads spanning at least 20 bases on each side of the junction, with a maximum of 2 mismatches.

(3) Linear back-spliced RNA detection: reads from the poly (A) enriched fraction were also analyzed with the CIRCexplorer2 characterization pipeline to detect back-spliced junctions and aligned to Gencode GRCh38 using TopHat-Fusion.

Gene count (raw reads) datasets were analyzed with the DESeq2 Bioconductor package (v1.18.1) [28]. Gene expression level comparisons (as log2-fold change estimation of DESeq2 normalized counts) were performed using the Wald test statistics as implemented in the DESeq2. The data for linear mRNA reads (Appendix A), circRNA reads (Appendix A) and linear back-spliced reads (Appendix A) of the Daoy (Appendix A) and HEPM (Appendix A) cells are presented. The Cufflink assembled circRNAs are included in the Appendix A and were used for generation of the DESeq2 normalized count datasets, but were excluded from further analysis.

For the volcano plots, genes having more than two samples with normalized reads lower than 1 and mean reads across all samples lower than 2 (cutoff = 2) were filtered out. For the circRNAs in the Daoy cells volcano plots, the upregulated and downregulated RNAs are depicted using less stringent criteria relative to the other volcano plots (|log2 fold change| > 0.5, *p*_value < 0.05 instead of |log2 fold change| > 1, *p*_adj < 0.05), as implementation of the stringent criteria resulted in no regulated RNAs.

### 2.7. Validation of Back-Spliced Junctions by Sanger Sequencing

All RNAs with back-spliced junctions that were used experimentally were first validated by Sanger sequencing. For this, divergent PCR primers on the back-spliced exons were designed using the NCBI primer blast tool. cDNA was generated as described in the next section and PCR reactions were performed with Q5 Hot Start High-Fidelity DNA Polymerase (M0493S, NEB, Ipswich, MA, USA) using 1000 ng template cDNA, in a 50 uL reaction mix under the following conditions: 98 °C for 30 s, followed by 35 cycles of 98 °C for 10 s, 65 °C for 15 s, 72 °C for 30 s, and 72 °C for a 2 min final extension. PCR fragments were visualized with Gel Red Nucleic Acid Gel Stain (41003-T, Biotium, Fremont, CA, USA) in 2% agarose gel (35-1020, Agarose Universal, peqGOLD, VWR, Radnor, PA, USA). DNA bands corresponding to the length of the expected PCR products were purified with the QIAquick Gel Extraction Kit (28704, Qiagen) and the presence of the anticipated back-spliced junction was validated by Sanger sequencing (Eurofins Genomics, Ebersberg, Germany) (Appendix A). Later, the same primer pairs were used for qPCR quantifications of circular and back-spliced linear transcripts as described below.

### 2.8. cDNA Synthesis, Real-Time qPCR

RNA was quantified spectrophotometrically with Infinite 200 NanoQuant microplate reader (TECAN, Männedorf, Switzerland), and 1000 ng of RNA was used for cDNA synthesis. cDNA was generated with M-MLV RT reverse transcriptase (28025013, Invitrogen, Thermo Fisher Scientific, Waltham, MA, USA) using random N6 primers (S1230S, NEB, Ipswich, MA, USA). Real-time qPCR was performed using FastStart Universal SYBR Green Master (Rox) (Roche, Merck, Darmstadt, Germany) on a 7500 fast real-time PCR system (Applied Biosystems, Waltham, MA, USA), with primers to detect the linear or circRNA transcripts (Appendix A). The primers spanning two adjacent exons to avoid genomic DNA amplification were designed using the NCBI primer blast tool. All amplifications were run in triplicate and the fold change was normalized to the expression of the TBP housekeeping gene. All qPCR reactions were performed under the following conditions: 95 °C for 10 min, followed by 40 cycles of 95 °C for 10 s and 65 °C for 30 s. The relative expression was determined either with the 2^−ΔΔCt^ method [29], by subtracting the Ct value of the housekeeping gene from the Ct value of the interrogated transcripts (∆Ct), and normalized to the ∆Ct values obtained with the control treatment (∆∆Ct) or the 2^−ΔCt^ method, where the Ct values of the housekeeping gene is subtracted from the Ct value of the interrogated transcripts (∆Ct). Negative controls (no addition of reverse transcriptase) for each primer pair were also run to ensure no DNA contamination in each of the samples.

### 2.9. Validation of Linear Back-Spliced RNAs by qPCR

cDNA synthesis and qPCR were performed as described above. Moreover, cDNA synthesis was also performed using oligo(dT) primers (S1316S, NEB, Ipswich, MA, USA). In the reverse transcription reaction, while random hexamers prime both linear (poly (A)) and circular (non-poly (A)) RNAs, oligo(dT) can effectively prime only linear (poly (A)) RNAs.

### 2.10. siRNA and Plasmid Transfections

Specific siRNAs targeting the back-spliced junction of FKBP8, SMARCA5, GLIS1, BACH1, ZKSCAN1, CDYL, and OGDH circRNAs (Appendix A) and control siRNAs (SIC001, Sigma-Aldrich, Merck, Darmstadt, Germany) were used. We have designed three siRNAs for CDYL, BACH1, and GLIS1 circRNAs, and two for OGDH, SMARCA5, ZKSCAN1, and FKBP8 circRNAs using online tools (https://horizondiscovery.com/en/ordering-and-calculation-tools/sidesign-center, https://circinteractome.irp.nia.nih.gov/siRNA_design.html, accessed on 2 March 2020).

For plasmid transfections, the full-length FKBP8, SMARCA5, GLIS1, BACH1, ZKSCAN1, CDYL, and OGDH circRNAs were amplified and cloned into the circRNA overexpression vector pCD5-ciR (GS0105, Geneseed, Guangzhou, China), with the empty vector used as a negative control.

Daoy and UW-228 cells were seeded in 96-well, 24-well, or 12-well plates at about 60% confluency, grown overnight until 70–80% confluent and transfected with 1 pmol siRNAs in 96-well plates, 5 pmol siRNAs in 24-well plates for 72 h, and 1000 ng plasmids in 12-well plates for 48 h. Lipofectamine RNAiMAX transfection reagent (Invitrogen, Waltham, MA, USA) was used for siRNA, and Lipofectamin 3000 transfection reagent (Invitrogen, Waltham, MA, USA) for plasmid transfections according to the manufacturer’s instructions.

### 2.11. Cell Proliferation Assays

#### 2.11.1. EdU Incorporation Assay

Cells were seeded in 12-well or 24-well plates and transfected at 70% confluency with 5 pmol siRNAs or 1000 ng overexpression plasmids for 72 or 48 h, respectively, followed by 2 h of 10 μM EdU (5-ethynyl-2-deoxyuridine, Invitrogen, Waltham, MA, USA) incubation. The cells were treated with control siRNA or pCD5-ciR vector as the negative control. Incorporated EdU was detected by the click reaction according to the manufacturer’s instructions (C10425, Click-iT EdU Alexa Fluor^®^ 488 Flow Cytometry kit, Invitrogen, Waltham, MA, USA). For each treatment, 10,000 cells were analyzed on a CytoFLEX flow cytometer (Beckman Coulter, Indianapolis, IN, USA).

#### 2.11.2. WST-1 Viability Assay

In total, 3 × 10^4^ cells were seeded in 96-well plates and left to grow overnight until 80% confluent. Cells were transfected with 1 pmol siRNAs for 72 h. Cell metabolic activity was measured with the WST-1 reagent (05015944001, Roche, Switzerland) according to the manufacturer’s instructions and the fraction of metabolically active cells was quantified at 450 nm, with a reference wavelength of 690 nm (Infinite M200 PRO, TECAN, Switzerland).

### 2.12. Statistical and Bioinformatic Analysis

Experiments were performed independently at least three times, unless otherwise stated in the figure legends. For RNA Illumina-seq, four independent replicates from each treatment and control group were used. The results are generally represented as the mean value and standard error of the mean of independent experiments. A detailed description of the statistical analyses is given in the figure legends. Once the general assumptions of the specific statistical tests were met, a parametric or nonparametric analysis was applied. Statistical analyses of the qPCR assays were performed using GraphPadPrism v8.2.1 (GraphPad Software, San Diego, CA, USA), and RNA-seq data analyses was performed using R v3.6.0 (R Foundation for Statistical Computing, Vienna, Austria) and Python v3.7.6 (Python Software Foundation, Scotts Valley, CA, USA).

In short, RNA-seq data normalization and differential expression analysis was performed in the Deseq2 library (v1.26.0, R) and visualized as volcano plots using the ggplot2 library (v3.3.0, R). In Appendix A–D, heatmaps of the log-transformed DESeq2 counts were generated using pheatmap library (R). Appendix A, Kruskal’s non-metric multi-dimensional scaling analysis (MDS), was performed using the isoMDS function (MASS library v7.3.51.6, R).The scatterplots were built with the seaborn.jointplot function (seaborn library v0.11.0, Python), which allows to study the relationship between the two variables plotted (circular and linear DESeq2 counts). Log2(x + 1) transformation was applied to the circular and linear counts. The central chart displays their correlation and the marginal charts at the top and at the right show the distribution of each variable with the histogram and kernel density plots. A regression line is drawn as a simple linear model fit between the variables. As circular datasets are not normally distributed, a Spearman nonparametric rank correlation coefficient and associated *p*-value between the two variables is annotated on the graphs using the scipy.stats.spearmanr function (scipy library v1.5.2, Python), which assesses how well the relationship between these variables can be described using a monotonic function. A correlation of −1 or +1 imply a monotonic relationship, and 0 imply no correlation.

The Venn diagrams were built with the venn2 functions in the matplotlib-venn package (Python). The heatmaps were built with the seaborn.clustermap function (seaborn library v0.11.0, Python). Data were normalized by rows (representing gene transcripts) and presented as z-scores, z = (x-mean)/std, with x being the circular to linear ratio of a transcript (circular or linear counts in Appendix A) in each of the control or treated sample; therefore, the values in each row will have the mean ratio of the row subtracted, then divided by the standard deviation of the row. This ensures that each row has a mean z-score of 0 and a variance of 1.

## 3. Results

### 3.1. Detection of Abundantly Expressed circRNAs in the Context of Hedgehog Signaling Upregulation

#### 3.1.1. Hedgehog Signaling Activation in Daoy and HEPM Cells

Most human cell lines lose their capacity to transduce the Hedgehog signal; however, the Daoy, a SHH subtype of medulloblastoma [30,31], and the non-cancerous HEPM (human embryonic palatal mesenchyme) cells maintain it. Thus, we activated Hedgehog signaling in Daoy and HEPM cell lines via administration of purified SHH ligand or the Smoothened agonist SAG, which activates the central signaling molecule of the pathway. qPCR analysis of key markers of the Hedgehog pathway activity, the GLI1 transcription factor, and the typical Hedgehog target gene HHIP demonstrated an over 10-fold upregulation of GLI1 (SAG/SHH treatments) and an over 20-fold upregulation of HHIP (SHH treatment) in Daoy and HEPM cells (Figure 1A). Following the verification of a major upregulation of Hedgehog signaling, the same RNA preparations were subjected to Illumina RNA-seq.

#### 3.1.2. RNA-Seq: Detection of Linear, Circular, and Linear Back-Spliced Transcripts

Total RNA from four replicates of each treatment (SAG/SHH) or of control groups was divided into two fractions. The first fraction was subjected to standard RNA-seq analysis for detection of linear mRNAs. The second fraction was treated with the exonuclease RNase R, which digests linear but not circRNAs, and then subjected to specialized RNA-seq analysis for detection of back-spliced exon junctions via the CIRCexplorer2 pipeline [27]. Moreover, the CIRCexplorer2 pipeline was also implemented in the RNA-seq data from the first fraction, allowing the identification of back-spliced junctions in linear mRNAs (see Materials and Methods, Section 2.5 and Section 2.6, for details).

#### 3.1.3. Linear RNA-Seq Data Analysis. Confirmation of Hedgehog Pathway Activation in Daoy and HEPM Cells

The major Hedgehog signaling activation observed in the qPCR assays (Figure 1A) was confirmed in the linear RNA-seq analysis, as demonstrated in the volcano plots, with differentially expressed (DE) genes defined as mRNA transcripts having at least a 2-fold change (|log2 fold change| > 1, *p*_adj < 0.05) in expression level of DESeq2 normalized reads in treatment versus control groups. Thus, GLI1 and HHIP, as well as the additional marker of pathway activity, PTCH1, were among the most upregulated genes upon SAG/SHH treatment both in Daoy (Figure 1B, Appendix A) and HEPM (Figure 1C, Appendix A) cellular contexts.

#### 3.1.4. circRNA-Seq Data Analysis. Differentially Expressed circRNAs in SAG and SHH Treated Daoy and HEPM Cells

Next, DE circRNAs were defined as having at least a 2-fold change in the expression level in the SAG/SHH treatment group versus the corresponding control (|log2 fold change| > 1, *p*_adj < 0.05) in HEPM cells, and at least a 1.5-fold change (|log2 fold change| > 0.5, *p* < 0.05) in Daoy cells. Moreover, implementation of a mean read count cutoff of 40 selected the most abundant DE circRNAs. These were FAM13B, RARS, ATXN10, BACH1, and OGDH in Daoy (Figure 1D, Appendix A) and CBFA2T2, FKBP8, FGFR1, MARK4, RTN4, and LRBA in HEPM (Figure 1E, Appendix A) cells. In addition, the 5 most abundantly expressed circRNAs from Daoy that were also among the top 50 in HEPM (Appendix A), and the top 5 from HEPM that were also among the top 50 in Daoy (Appendix A) cells were selected. These were CDYL, UBXN7, HIPK3, ZKSCAN1, ASXL1 and LPAR1, HIPK3, ASXL1, CORO1C, and RHOBTB3, respectively. Moreover, four circRNAs among the top 50, which originate from genes relating to Hedgehog signaling, SMO, GLIS1, GLIS2 and GLIS3 were also selected. These together with SMARCA5, a top 20 circRNA in both Daoy and HEPM (Appendix A), resulted in a list of 24 unique circRNAs. Using divergent primers to PCR amplify the back-spliced junction of these transcripts (Appendix A), all but one (RTN4) were validated by Sanger sequencing (Table 1, Appendix A). To our surprise, qPCR analysis of the selected DE circRNAs did not confirm the statistically significant but minor (2- or 1.5-fold) expression differences between the SAG/SHH treatment compared to the control (Appendix A).

#### 3.1.5. Linear Back-Spliced RNA-Seq Data Analysis and qPCR Validations

Moreover, linear poly (A) enriched RNA-seq reads from Daoy and HEPM cells were also analyzed with the CIRCexplorer2 (TopHat-Fusion) pipeline to detect linear poly (A) back-spliced transcripts. In both cell lines these transcripts were much less abundant than the circular back-spliced RNAs. There were only six linear back-spliced transcripts detected in HEPM and only one in Daoy cells, with a mean read count cutoff ≥ 2 across the samples, and none being DE in the SAG or SHH treatment (Appendix A). COL1A2, the most abundant transcript in HEPM, and TTC21B from Daoy were selected for further validations. Although both transcripts were successfully PCR amplified from Daoy cell RNA (COL1A2 is also expressed in Daoy but not as high as in HEPM (Appendix A) and validated with Sanger sequencing (Appendix A), only COL1A2 had a reliable Ct threshold range in the qPCR analysis.

To confirm the linear nature of the COL1A2 back-spliced RNA by an independent method, the following approach was used: first, cDNA from Daoy cells was prepared with either a random hexamer or oligo(dT) primers; then, the RNA expression of two representative circRNAs, CDYL and HIPK3, was compared to that of the COL1A2 back-spliced RNA in qPCR assays. As anticipated, while the COL1A2 RNA was detected with comparable efficacy both in the random hexamer and oligo (dT) cDNA preparations, the circular CDYL and HIPK3 RNAs were about 10-fold less effectively detectable in the oligo(dT) than in the random hexamer cDNA preparations (Figure 1F). This is consistent with COL1A2 being a linear poly (A) mRNA, with a repetition of the “back-spliced” exon, exon 32, which likely results from intermolecular splicing of independent pre-mRNAs [32,33,34].

### 3.2. Circular to Linear Ratio Analysis of RNA-Seq Data

To analyze the circular to linear RNA ratios in the RNA-seq data from Daoy and HEPM cell lines, first, circRNAs were evaluated for possible correlation with their linear counterparts. Although circRNA reads are inherently skewed towards low reads, as circRNA-seq data are mostly comprised of very lowly expressed circles, the low Spearman rank correlation coefficients (0.024, *p* < 0.05 for Daoy and 0.038, *p* < 0.05) indicate an overall poor correlation between the circular and linear reads from the same gene (Figure 2A,B). This suggests that the mechanisms generating RNA circles are distinct to the ones involved in mRNA synthesis.

As most of the circular transcripts had very low expression, we filtered out the very lowly expressed circRNAs with a mean read count < 2 (cutoff = 2). At this cutoff threshold, 3553 circRNAs in Daoy and 2440 circRNAs in HEPM cells were detected (Appendix A), of which 1724 were common between the two cell lines (Appendix A).

Implementation of a more stringent threshold (cutoff = 40) selected relatively highly expressed circRNAs (Appendix A) and eliminated circular transcripts with very low (log2 (circ/lin) < −10 for Daoy and log2 (circ/lin) < −12 for HEPM) circular/linear ratios (Figure 2D,E). This selection cutoff resulted in 83 unique transcripts in Daoy and 64 unique transcripts in HEPM cells, of which 31 were common between the two cell lines (Figure 2C).

Interesting to note is that for these relatively abundant circRNAs, the z-normalized circular to linear ratios are clustering differently in the SAG/SHH treatment groups compared to the respective control treatments (Figure 2F,G and Appendix A), supporting the notion that circular and linear splicing is not concordant and likely independently regulated (Appendix A). In HEPM cells, the circular/linear ratios are noticeably lower (predominance of dark blue to yellow) in the SAG and SHH treatments versus the control treatment (Figure 2G and Appendix A).

Interesting to note is that among the selected and validated circRNAs (Table 1), a substantial number are above the 80th percentile of the circular read scores and below the 90th percentile of the linear read scores (Appendix A). Specifically, in Daoy cells, CDYL has the highest circular counts and at the same time among the highest circular/linear ratios, which is minimally affected by the SAG/SHH treatment (Appendix A). On the other hand, in HEPM cells, LPAR1 has the highest circular counts and at the same time among the highest circular/linear ratios, which is reduced following the SAG/SHH treatment (Appendix A).

### 3.3. Differentially Expressed circRNAs in SHH Subtype Medulloblastoma (Daoy, UW-228 Cell Lines and Tumor Sample) and Normal Cerebellum

To further expand our circRNA-seq analysis we took advantage of the Rybak-Wolf et al. circRNA-seq dataset from human cerebellum [16]. First, from the 50 most abundantly expressed circRNAs in Daoy (Appendix A), we removed the duplicated entries resulting from distinct Ensembl transcripts encompassing the same back-spliced junction and the Cufflink assembled circRNAs, which may originate from non-referenced genes. This reduced the number to 37 circRNAs, 33 of which had a circRNA ID in the circBase database (http://www.circbase.org/, accessed on 24 October 2019). These 33 circRNAs were compared to the Rybak-Wolf et al. dataset, and circRNAs with a Daoy/cerebellum fold change > 2 or <0.5 were selected (Appendix A). This resulted in a list of nine circRNAs, ARGHAP12, ATXN10-2, CCDC134, CDYL, GAS2, GLIS3, RHOBTB3, RSRC1, and SFMBT2, three of which, CDYL, GLIS3, and RHOBTB3, were present in the initially validated 23 circRNA list (Table 1). The six novel circRNAs were all validated by Sanger sequencing (Appendix A), expanding the list to 29 circRNAs (Table 1). A total of 23 circRNAs from this validated list were found to have good performance in qPCR assays in Daoy and UW-228 cells, another medulloblastoma cell line of the SHH subtype (Table 1) [31].

Next, we analyzed the expression of these 23 circRNAs in medulloblastoma (Daoy, UW-228 medulloblastoma cell lines, and a medulloblastoma tumor sample of the SHH subtype) and in normal cerebellum (three independent samples). Seventeen circRNAs were deregulated in medulloblastoma compared to normal cerebellum with statistical significance. In fact, all 17 were expressed at lower levels in the tumor (Figure 3A), in line with the claims that circRNA abundance negatively correlates with proliferation [35]. Four of the deregulated circRNAs, FGFR1, GLIS2, LRBA, and SFBMT2, had relatively high qPCR cycle threshold (Ct) values, indicating a very low abundance in the samples and poor reliability in the qPCR assay. The remaining 13 circRNAs with qPCR Ct values < 30 in Daoy cells were selected for further analysis.

We then examined the expression in the medulloblastoma and cerebellar samples of linear mRNAs originating from the same gene as these 13 circRNAs. Importantly, for seven of these mRNAs there was no statistically significant difference in medulloblastoma versus cerebellar expression, while for the remaining mRNAs the same pattern as for the circRNAs, i.e., a lower expression in medulloblastoma in comparison to cerebellum, was observed (Figure 3B). The seven circRNAs, FKBP8, SMARCA5, GLIS1, BACH1, ZKSCAN1, CDYL, and OGDH, with a reduced expression in medulloblastoma versus cerebellum, while their corresponding linear mRNAs do not follow this change of expression pattern (Figure 3C,D), were selected for further analysis. The differential splicing events that act on these circRNAs relative to their linear mRNAs support the possibility for a role in medulloblastoma development.

### 3.4. Effect of Depletion and Overexpression of the Seven Selected circRNAs on Cell Proliferation in Medulloblastoma Cell Lines

First the efficacy of siRNAs targeting the back-spliced junction of the FKBP8, SMARCA5, GLIS1, BACH1, ZKSCAN1, CDYL, and OGDH circRNAs was determined. The results indicated that in Daoy and UW-228 medulloblastoma cells at least one siRNA for each of the seven circRNAs could specifically downregulate the circRNA without affecting the corresponding linear mRNA (Appendix A). However, the depletion of these seven DE circRNAs did not elicit consistent changes in Daoy or UW-228 cell proliferation, as measured by EdU incorporation (Appendix A) or the WST-1 viability assay (Appendix A). In fact, it appears that proliferation tends to be reduced, while it would be expected that depletion of the downregulated RNA circles would increase cellular growth.

We further generated constructs to overexpress these seven circRNAs via the pCD5-ciR vector. In total, five out of seven constructs (CDYL, BACH1, GLIS1, SMARCA5, and ZKSCAN1) were successfully made and validated by Sanger sequencing (Appendix A). Overexpression of these constructs was verified by qPCR assays (fold change 2.8 to 25.4) (Appendix A). However, circRNA overexpression did not significantly affect cell proliferation (Appendix A); in fact, and in contrast to expectations, a trend towards increased cellular growth was observed. Collectively, these results indicate that depletion or overexpression of these deregulated circRNAs has minimal impact on cell proliferation in the two medulloblastoma cell lines analyzed.

## 4. Discussion

We have identified the circRNA transcriptome of two human cell lines, the SHH medulloblastoma Daoy and the embryonic palatal mesenchyme HEPM cells, in the context of Hedgehog signaling activation. Activation of the pathway, elicited by treatment with the purified SHH ligand or the small molecule SAG, an agonist of the Smoothened co-receptor, was validated in qPCR assays by the upregulation of the robust gene markers of signaling activity, GLI1 and HHIP (Figure 1A).

Next, total RNA from treated and untreated cells was divided into two fractions to enrich for either linear mRNAs (poly (A) selection) or circRNAs (ribosomal RNA depletion and RNase R treatment) and subjected to Illumina RNA-seq. The Hedgehog pathway upregulation seen in the qPCR assays (Figure 1A) was comprehensively confirmed in the RNA-seq analysis of the first fraction (mRNA detection) (Figure 1B,C). The key markers of the Hedgehog pathway activation, GLI1, HHIP, and PTCH1, were among the most upregulated genes upon SHH/SAG treatment of both Daoy and HEPM cells.

The RNA-seq analysis of the second fraction, centering on the detection of back-spliced exon junctions, identified five and six abundantly expressed circRNAs, which were apparently differentially expressed in Daoy and HEPM cells, respectively (Figure 1D,E). Interestingly, all these DE circles, with the exception of one, were downregulated in the context of Hedgehog signaling activation both in Daoy and HEPM cells. Surprisingly, the qPCR analysis of the DE circRNAs did not show consistent differences in expression compared to the control. This might be due to the possibility that at smaller expression differences (2-fold or 1.5-fold), the RNA-seq data reflecting absolute counts might be more sensitive than a qPCR assay, which represents copy numbers of a particular transcript relative to a linear housekeeping gene. In addition, circRNAs are considerably more stable, with a half-life of 24–48 h, compared to linear mRNAs, which are usually degraded within few hours [36,37]. Thus, significant but small changes of circRNAs will be more difficult to detect within the 48–72 h timeframe of SAG/SHH treatment. This may be a possible explanation as to why fewer DE circRNAs were detected compared to DE mRNAs. In fact, it has been previously reported that unlike the rapid changes exhibited by mRNAs and miRNAs, the expression levels of circRNAs in MCF10A mammary cells were minimally altered following a 4-h stimulation by epidermal growth factor [38], while a 21-day TGF-β treatment of epithelial cells resulted in major circRNA changes [39].

In addition to circRNAs, we have also identified back-spliced exon junctions in the poly (A) enriched RNA fraction. However, these linear back-spliced RNAs were less abundant compared to circular back-spliced RNAs. Importantly, we show that while the linear back-spliced COL1A2 was detected with a similar efficiency in cDNA reverse transcribed with either a random hexamer or oligo (dT) primers, highly abundant circRNAs, e.g., CDYL and HIPK3, were almost undetectable in the oligo (dT) primed cDNA (Figure 1F). Poly (A) RNA, i.e., mRNA, with a back-spliced exon junction is rationalized by the phenomenon of exon repetition, elicited by intermolecular splicing (trans-splicing) of independent transcripts, which is facilitated, as with circRNAs, by inverted repeats in the flanking introns [40]. Although a circRNA for COL1A2, with the same back-spliced junction as the poly (A) COL1A2 RNA, was also detected in the RNA-seq data, its expression was much lower (Appendix A). Consequently, it is highly unlikely that the finding of poly (A) back-spliced RNAs is an “artifact” of the presence of abundant circRNAs. Rather more plausible is a scenario where mechanisms similar to the ones promoting circRNA expression, e.g., inverted repeats in flanking introns, may, depending on context, also promote intermolecular splicing of linear transcripts.

In general, both in Daoy and HEPM cells the expression of circRNAs was lower than that of linear mRNAs originating from the same parental gene. The correlation of circRNA and corresponding linear mRNA reads was also very low (Spearman correlation coefficients 0.024 for Daoy cells and 0.038 for HEPM cells), which is consistent with a previous study [24], and indicative of distinct mechanisms involved in back-splicing compared to forward-splicing.

Recently, the increased interest in circRNAs has led to the development of over 20 circRNA databases, including CircBase [41], CircAtlas [42], and Circpedia [43]. CircBase is a collection of nine circRNA datasets from previously published studies, including the human normal cerebellum dataset [16], which was compared to our data from the Daoy medulloblastoma cell line (Appendix A), and expanded the initially selected circRNA list (Table 1). The fact that this comparison was, in general, not in line with the qPCR data of Figure 3A is likely to relate with the different cerebellar samples in the two analyses. The Rybak-Wolf dataset uses two fetal samples of 19 and 37 weeks, while we have used three adult samples with an age ranging from 21 to 29 years.

The expression of the 23 validated circRNAs that performed well in qPCR assays was analyzed in medulloblastoma and normal cerebellum (Figure 3A). We used the three samples from cerebellum mentioned above as, to our knowledge, no human cerebellar cell lines are available. To minimize bias from comparing expression in medulloblastoma cell lines versus cerebellum, we also included a SHH medulloblastoma tumor sample. Interestingly, most of the circRNAs were significantly downregulated in medulloblastoma compared to cerebellum, with none being upregulated. This is consistent with previous studies, as circRNAs appear to be preferentially downregulated in cancers [35,44], including medulloblastoma [45]. In fact, out of the 33 DE circRNAs reported by Lv et al., only three were upregulated, while the remaining 30 were downregulated in medulloblastoma compared to normal cerebellum. However, these DE circRNAs were different to the ones from our study, possibly because Lv et al. did not focus on the SHH subtype of medulloblastoma. Next, we examined the expression of mRNAs originating from the same genes as the circRNAs that were downregulated in medulloblastoma. For seven of the downregulated circRNAs (ranging from 3-fold in CDYL, up to 22-fold in FKBP8) (Figure 3A), the expression of their linear counterparts remained unchanged (Figure 3B). Among these seven circRNAs, BACH1 and FKBP8, based on the RNA-seq data, are also downregulated following Hedgehog pathway activation (Figure 1D,E and Appendix A), while no expression differences are seen in the corresponding mRNAs (Appendix A). These findings highlight that the regulation of expression of circRNAs can be independent to the expression of their corresponding mRNAs. An independent regulation of a biological process, i.e., back-splicing, may have difficulties to be seen as a simple by-product of a more generalized process, i.e., RNA splicing, and therefore could be associated with a functional impact.

With this line of reasoning, we explored the possibility of functional implications of these seven circRNAs in two medulloblastoma cell lines, Daoy and UW-228. To our surprise, neither siRNA knockdown nor transient overexpression of these circRNAs elicited consistent and significant changes in Daoy or UW-228 cell proliferation. In fact, and in contrast to expectations, depletion appeared to reduce and overexpression tended to increase proliferation. These findings may suggest that despite a relative abundance and a differential expression, considerable caution must be taken when addressing the functional roles of circRNAs in tumor cell growth [46]. However, it is also possible that these RNA circles may impact medulloblastoma cells in ways that our assays are unable to detect [47,48,49,50].

Of note, GLIS1 is expressed in Daoy but not in UW-228 cells, highlighting differences in the gene expression patterns of these two medulloblastoma cell lines. This is consistent with previous studies, which indicate that 58% of the proteins identified in Daoy were not detected in UW-228 [30,51]. Despite this, the outcomes of the Daoy and UW-228 analysis was rather similar.

## 5. Conclusions

Seven circRNAs with reduced expression in medulloblastoma versus cerebellum, while their corresponding mRNAs remained unchanged, have been identified. However, their depletion and/or overexpression had no significant impact on cell proliferation in Daoy and UW-228 medulloblastoma cells, based on two independent assays. Nevertheless, some of these circRNAs have established biological roles in other tumor cells, e.g., circCDYL facilitates the progression of cervical cancer by targeting the miR-211-5p/SOX4 axis [52] and inhibits colorectal cancer cell proliferation and migration via the miR-382-5p/PTEN axis [53]. It is possible that more specialized assays may reveal implications of the seven circRNAs in medulloblastoma growth, although a scenario of representing “passenger molecules”, with limited functional impact, cannot be excluded.

Finally, it should also be mentioned that the seven circRNAs were identified using a limited number of different tumor cells. Consequently, an analysis of a larger collection of medulloblastoma samples may provide a more robust list of deregulated circRNAs, which could clearly impact cerebellar tumor growth.

## Figures and Tables

**Figure 1 cancers-13-05138-f001:**
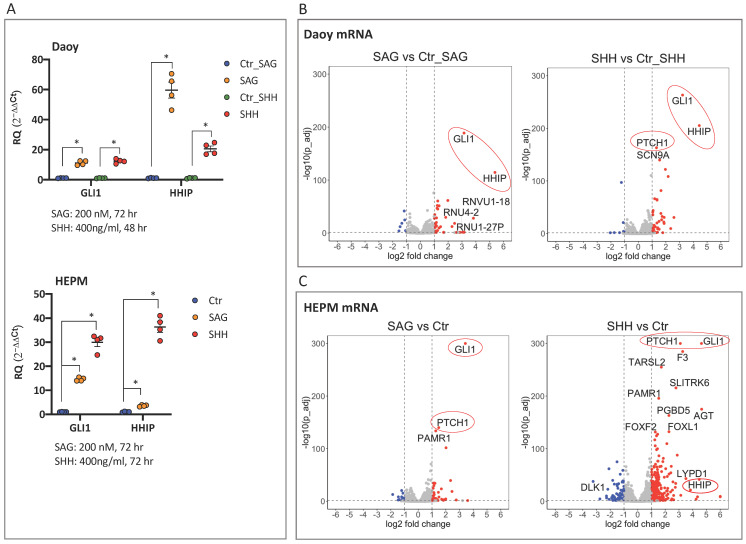
circRNAs regulated by Hedgehog signaling activation in human medulloblastoma Daoy and human embryonic palatal mesenchyme HEPM cell lines: (**A**) qPCR analysis of Hedgehog target gene expression. The relative expression (2^−ΔΔCt^ values), normalized to the housekeeping gene (TBP) and the control treatment (DMSO for SAG and PBS for SHH in Daoy cells, DMSO + PBS in HEPM cells), is presented. Error bars indicate the SEM of four independent replicate treatments. RQ denotes the relative quantification of the mRNA expression. Multiple t-test using Holm–Sidak’s method (*p* values computed while not assuming consistent SD) was applied to calculate the statistically significant differences (*: *p* < 0.05) between each treatment versus the control, highlighting the effectiveness of the Hedgehog pathway upregulation. (**B**–**E**) Illumina RNA-seq analysis of the same RNA samples depicted in (**A**). Volcano plots visualizing the differentially expressed linear mRNAs (**B**,**C**) or circRNAs (**D**,**E**) following SAG or SHH treatment in Daoy and HEPM cells. The upregulated Hedgehog target genes in (**B**,**C**) are highlighted in red circles. Cutoff thresholds are assessed with the DESeq2 method, where the Wald significance test is applied as (|log_2_ fold change| > 1, *p*_adj < 0.05) for linear mRNA expression in Daoy and HEPM as well as for circRNA expression in HEPM, and (|log_2_ fold change| > 0.5, *p* < 0.05) for circRNA expression in Daoy cells. Red and blue dots indicate upregulated and downregulated genes, respectively. Annotated are the mRNAs with mean read counts > 40 and (|log_2_ fold change| > 3 or −log_10_ (*p*_adj) > 130) (**B**,**C**) and the selected DE circRNAs (**D**,**E**). (**F**) Comparison of circular (CDYL and HIPK3) and linear (COL1A2) back-spliced RNA expression in cDNA preparations generated with random hexamers or oligo(dT) primers in Daoy cells. The expression of a given transcript first normalized to the housekeeping gene TBP (2^−ΔCt^ values) and then to the corresponding (2^−ΔCt^)-normalized expression in the random hexamers-primed cDNA is presented. RQ denotes the relative quantification of the RNA expression. Error bars indicate the SEM of three independent experiments. Note that while the linear COL1A2 RNA is detected with similar efficiency, both with random hexamers and oligo(dT) primers, the circular CDYL and HIPK3 RNAs are almost undetectable with oligo(dT) primers. Multiple t-test using Holm–Sidak’s method was applied to calculate the statistically significant differences (**: *p* < 0.01 and ***: *p* < 0.001) between the RNA expression in the random hexamer versus oligo(dT) primed cDNA preparations.

**Figure 2 cancers-13-05138-f002:**
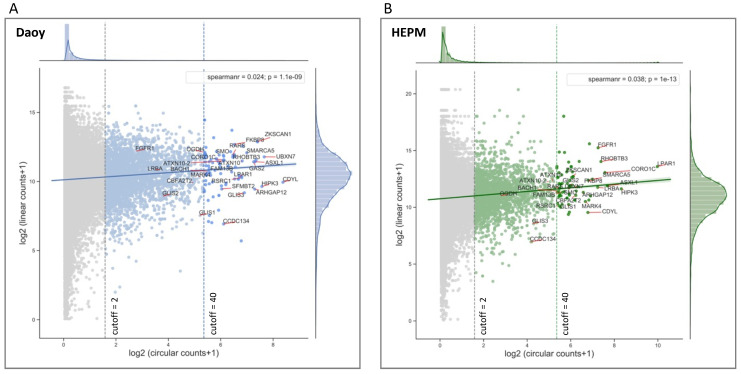
circRNA expression is not correlated with the respective linear mRNA expression. (**A**) Daoy and (**B**) HEPM cell Illumina RNA-seq DESeq2 normalized mean circular counts (as log_2_ (circular counts + 1) on the x axis) are plotted against their respective mean linear counts (as log_2_ (linear counts + 1) on the y axis) together with the regression fits between the circular and linear counts. Marginal plots at the top and at the right represent histogram and kernel density plots for the circular and respective linear counts. The low Spearman rank correlation coefficients indicate a poor relationship between the circular and linear RNA-seq reads from the same gene. The mean circular read cutoffs were implemented to differentiate the low (cutoff = 2) and the highly (cutoff = 40) expressed circRNAs. Selected genes (Table 1) are annotated in the plots. Note the relatively small fraction of genes that pass the mean circular read cutoff = 40. (**C**) Venn diagram representing the overlap of circRNAs expression in Daoy and HEPM cells at cutoff = 40. (**D,E**) Histogram and kernel density (kde) plots (hist) of the log_2_ (circular/linear) ratios for the RNA circles at low (cutoff = 2) and high (cutoff = 40) read thresholds in Daoy (**D**) and HEPM (**E**) cells. (**F**,**G**) Heatmaps with hierarchical clustering based on circular/linear ratios of DESeq2 reads at cutoff = 40, by treatment group (Control, SAG, or SHH) in Daoy (**F**) and HEPM (**G**) cells. Shown in each row are the normalized z-scores (see Materials and Methods) for each gene. Yellow indicates higher circular/linear ratios, while blue lower. Selected genes (Table 1) are highlighted in red.

**Figure 3 cancers-13-05138-f003:**
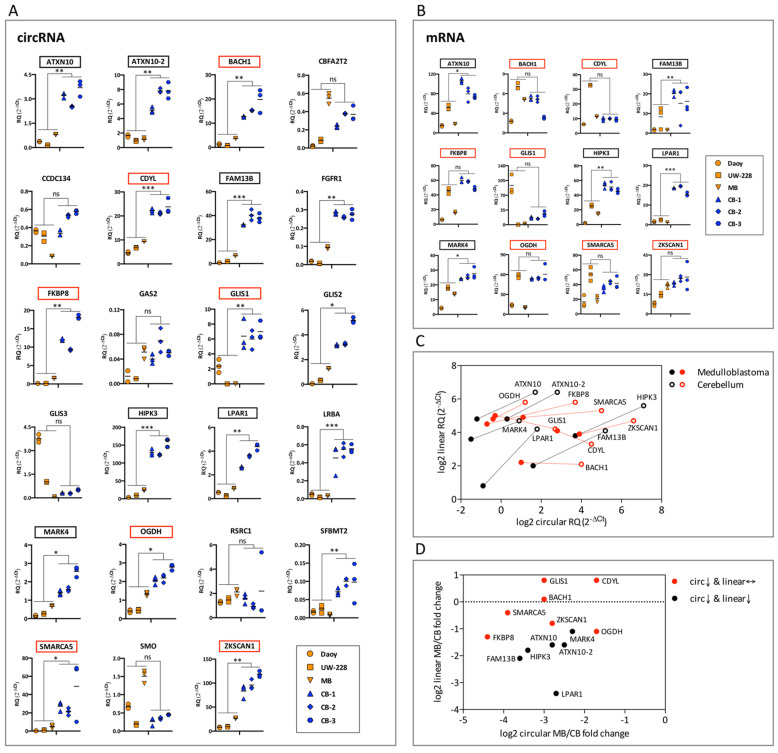
Differentially expressed circRNAs in medulloblastoma and cerebellum. (**A**) qPCR analysis of circRNAs in medulloblastoma (Daoy, UW-228, and a SHH medulloblastoma human sample) versus cerebellum (three independent human samples). RNA expression is presented as relative expression (2^−ΔCt^ values), normalized to the housekeeping gene (TBP). Short lines indicate the mean of the technical replicates per sample. RQ denotes the relative quantification of the RNA expression. Multiple t-test using the Benjamini and Hochberg method (*p* values computed while not assuming consistent SD) was applied to calculate statistically significant differences (*: *p* < 0.05, **: *p* < 0.01 and ***: *p* < 0.001) between the medulloblastoma versus the cerebellar samples. The selected 13 circRNAs are highlighted in red or black boxes depending on the expression of their corresponding mRNAs. (**B**) qPCR analysis in medulloblastoma versus cerebellum of the 12 mRNAs originating from the same gene as the 13 selected circRNAs. The analysis was performed as in (**A**). Red boxes highlight mRNAs that remain unchanged in medulloblastoma, while black boxes mRNAs were downregulated in medulloblastoma. (**C**) Correlation of expression in medulloblastoma and in cerebellum of the selected 13 circRNAs with their corresponding linear mRNAs. The log_2_ of the mean expression (2^−ΔCt^ values) in medulloblastoma (filled dots) and in cerebellum (hollow dots) of circRNAs (x axis) and the linear mRNAs originating from the same gene (y axis) is presented. Red dots highlight circRNAs significantly downregulated in medulloblastoma, while their corresponding mRNAs remain unchanged, and black dots circRNAs whose corresponding mRNAs are also downregulated in medulloblastoma. Lines connect the individual gene expression in cerebellum and medulloblastoma. (**D**) Correlation of the fold change in expression in medulloblastoma versus cerebellum of the selected 13 circRNAs with their corresponding linear mRNAs. The log_2_-fold change of the mean expression (2^−ΔCt^ values) in medulloblastoma versus cerebellum (MB/CB) of circRNAs (x axis) and the linear mRNAs originating from the same gene (y axis) is presented. Red and black dots are as in panel (**C**).

**Table 1 cancers-13-05138-t001:** Selected and Sanger sequencing validated circRNAs.

	circRNA		circRNA ID	Detected by qPCR
1	ARGHAP12		hsa_circ_0000231	
2	ASXL1	*	hsa_circ_0001136	
3	ATXN10	*	hsa_circ_0001247	√
4	ATXN10-2		hsa_circ_0001246	√
5	BACH1	*	hsa_circ_0001181	√
6	CBFA2T2	*	hsa_circ_0003426	√
7	CCDC134		hsa_circ_0008806	√
8	CDYL	*	hsa_circ_0008285	√
9	CORO1C	*	NA	
10	FAM13B	*	hsa_circ_0001535	√
11	FGFR1	*	hsa_circ_0008016	√
12	FKBP8	*	hsa_circ_0000915	√
13	GAS2		hsa_circ_0021516	√
14	GLIS1	*	hsa_circ_0002079	√
15	GLIS2	*	hsa_circ_0005692	√
16	GLIS3	*	hsa_circ_0001246	√
17	HIPK3	*	hsa_circ_0000284	√
18	LPAR1	*	hsa_circ_0087960	√
19	LRBA	*	hsa_circ_0006867	√
20	MARK4	*	hsa_circ_0000940	√
21	OGDH	*	hsa_circ_0003340	√
22	RARS	*	NA	
23	RHOBTB3	*	hsa_circ_0007444	
24	RSRC1		hsa_circ_0001355	√
25	SFMBT2		hsa_circ_0000211	√
26	SMARCA5	*	hsa_circ_0001445	√
27	SMO	*	hsa_circ_0001742	√
28	UBXN7	*	hsa_circ_0001380	
29	ZKSCAN1	*	hsa_circ_0001727	√

* Initially selected 23 circRNAs; NA, not available.

## Data Availability

The raw RNA-seq data are available at NCBI, GEO accession number: GSE183194.

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
