# Peer review of "Circular RNAs in Hedgehog Signaling Activation and Hedgehog-Mediated Medulloblastoma Tumors"

_cancers, 2021, doi:10.3390/cancers13205138_

Round 1

Reviewer 1 Report

Minor

In the introduction section it seems that are alterations in hedgehog signalling responsible for the 25% of human cancer deaths. Is that so?? Otherwise the sentence should be rephrased.

Figure 1 should be formatted

Major

The study has been carried out comparing data obtained from cell lines to control cerebellum tissues. There is a brief mention to this in the conclusion section, but it should be further discussed in the discussion section. Indeed, the authors should comment on the limitations and potential bias (and their implication for the relevance of the results they find) of comparing immortalized cell lines with normal tissue. They should also discuss how the bias of comparing normal cerebellar RNA samples to that from a single cell type.

The authors found 33 circRNA transcripts had uniquely annotated circRNA IDs in the circBase database from the top-50 circRNAs. What could the relevance of the remaining 17 be? The authors should discuss this.

Also, the paragraph on page 12 (lines 439-452) is a bit hard to understand and should be rephrased.

Why did the authors decide to focus only on those 7 circRNAs?

On Figure 3 the authors compare circRNAs from medulloblastoma samples (2 cell lines and a patient) and cerebellar regular samples. The authors do not mention the differences they find within these subgroups, which are evident for some circRNAs. In this line, they select some circRNAs based on their statistical relevance, while discarding other circRNAs which are also statistically different. The criteria to select ones and not others should be better explained.

In this line, it is not clear what results derive from the comparison of patient derived SHH subtype medulloblastoma tumor tissue sample. Why did the authors focus on the results obtained in the cell lines?

Are there any public repositories with RNAseq data from medulloblastoma patients that the authors could use to increase their human sample?

The authors explain the relevance of Hedgehog signaling on medulloblastoma development, and identify alterations in circRNAs after Hedgehog signaling stimulation. However, how relevant is upregulation of this pathway for medulloblastoma biology? Is it essential? The RNAs are determined in medulloblastoma cell lines, and the levels are highly increased after stimulation. I guess the question is what are those circRNAs telling us about medulloblastoma?

The biology and relevance of Circular RNAs have gained interest in the last few years. However, the link between the relevance of a differential expression between circRNAs and mRNAs remains elusive. The roles of circRNAs go beyond cell proliferation (for a recent review see https://doi.org/10.1016/j.gendis.2020.07.012). the authors don't find an effect of the downregulation of circRNAs in the proliferation of their cell lines, but that is only one potential effect of circ RNAs. They should further comment on this.

Reviewer 2 Report

An important piece of information about the role of circular RNAs in Hedgehog signaling and Hedgehog-mediated medulloblastoma development has been provided by the authors. Using appropriate methods and techniques authors were able to identify 7 circRNAs, with reduced expression in medulloblastoma versus cerebellum while their corresponding mRNAs remained unchanged. Unlike other studies showing that overexpression of circRNAs can inhibit tumor cell growth, the depletion of these 7 circRNAs, as well as their overexpression didn’t have a significant impact on medulloblastoma cells proliferation. Although circRNAs’ role in cancer initiation and progression is widely studied recently, there is not yet a clear picture why for some cancers circRNAs can serve as an important anti-cancer “tool”, while for others have a promoting effect. Since this topic is of great interest, I highly recommend publishing the work in “Cancers”.

I have only some minor suggestions for figures. Some of them are very crowded like Fig2(A, B), most of the information is already in excel files anyways.

Font size is very small in some figures, it is hard to read, especially Fig.1 (B, C).

In Suppl. file S6 (Sanger-seq validations) - there are handwritings in the graphs: p.3 (ASXL1); p.24 (RHOBTB3); p.28 (SMO); p.31 (COL1A2)

Round 2

Reviewer 1 Report

Thank you for addressing all my questions. I think that the quality of the paper has increased, and is now easier to understand after yout modifications.

Author Response

We thank the reviewer for finding that the modifications implemented, in-line with the review report, increase the quality of the paper and make it easier to understand.